# Genome-Wide Identification and Expression Profile Analysis of the NADPH Oxidase Gene Family in *Avena sativa* L.

**DOI:** 10.3390/ijms26062576

**Published:** 2025-03-13

**Authors:** Qingxue Jiang, Xinyue Zhou, Jun Tang, Dengxia Yi, Lin Ma, Xuemin Wang

**Affiliations:** Institute of Animal Sciences, Chinese Academy of Agricultural Sciences, Beijing 100193, China; 18511590877@163.com (Q.J.); zhouxinyuenmg@163.com (X.Z.);

**Keywords:** NADPH oxidase, gene family, drought stress, salt stress, expression analysis

## Abstract

The plant respiratory burst oxidase homologs (*RBOHs*) are crucial enzymes responsible for the production of reactive oxygen species (ROS) in plants, playing a pivotal role in regulating various aspects of plant growth, development, and stress responses. While *RBOH* family members have been identified across a wide range of plant species, the functions and characteristics of the *RBOH* gene family in oats remain poorly understood. In this study, 35 members of the *RBOH* gene family in the oat genome were identified using bioinformatics approaches. Conserved motif and gene structure analyses revealed that most *AsRBOH* genes contain Motif4 and Motif5. Phylogenetic tree analysis demonstrated that the *AsRBOHs* can be classified into five distinct subfamilies. Synteny analysis indicated that *AsRBOHs* share the highest number of syntenic gene pairs with wheat. Additionally, cis-regulatory element analysis identified several elements associated with drought and hypoxia-specific responses in *AsRBOHs*. Expression analysis using qRT-PCR showed that 28 *AsRBOH* genes were upregulated under drought stress, while 18 were downregulated under salt stress. Notably, the genes *7DG1382190* and *7AG1225850* were found to be involved in both drought and salt stress responses. In conclusion, these findings provide a valuable foundation for future functional studies of the *AsRBOH* gene family in oats, offering insights that could contribute to the improvement and innovation of oat varieties and germplasm.

## 1. Introduction

NADPH oxidase, formally known as nicotinamide adenine dinucleotide phosphate oxidase, is a redox enzyme that catalyzes the generation of superoxide anions (O_2_^−^) from extracellular oxygen, utilizing NADPH as the electron donor in the cytoplasm [1]. Initially discovered in the phagocytes of the human immune system, NADPH oxidase is primarily found in the neutrophils of mammals, plant cells, and human neutrophils [2]. The enzyme consists of a core complex, which includes the subunits p40phox, p47phox, p67phox, p22phox, and gp91phox, along with two low-molecular-weight guanine nucleotide-binding proteins, Rac2 and Rap1A [3]. In plants, NADPH oxidase is referred to as respiratory burst oxidase homologs (*RBOHs*), which are homologous to the macrophage NADPH oxidase subunit gp91phox. *RBOHs* typically contain six conserved transmembrane domains and feature several key structural elements: a C-terminal hydrophilic domain for FAD and NADPH binding, two heme groups, and two N-terminal Ca^2^⁺-binding EF-hand domains [4,5,6]. In this process, NADPH acts as the electron donor, transferring electrons to extracellular O_2_^−^, which is subsequently reduced through the FAD and heme groups [7]. NADPH oxidase is predominantly localized to the plasma membrane, where it catalyzes the production of extracellular reactive oxygen species (ROS), which represent a major source of ROS in plants.

Reactive oxygen species (ROS) play a pivotal role in plant signal transduction, mediating processes such as growth and development, responses to biotic and abiotic stresses, and programmed cell death [8]. Recent studies have demonstrated that, under stress conditions, plant cells can sense and trigger adaptive responses through ROS signaling, a crucial mechanism for stress tolerance. NADPH oxidase, a key enzyme involved in ROS production, regulates the generation of ROS in response to various environmental and pathogenic stresses, thereby influencing plant growth and development [9,10,11,12]. To date, *RBOH* genes have been identified in several plant species, including *Arabidopsis thaliana*, *Oryza sativa*, and *Chenopodium quinoa* willd [13,14,15]. These genes typically exist as gene families; for instance, *O. sativa* contains nine *OsRBOH* genes [16], *A. thaliana* has ten *AtRBOH* genes, and *C. quinoa* harbors nine *CqRBOH* genes (*CqRBOH1*–*CqRBOH9*) [15]. Members of the *Arabidopsis AtRBOH* gene family, particularly *AtRBOHD* and *AtRBOHF* [17], have been implicated in ROS production during pathogen infection [18,19]. In *Nicotiana tabacum* (tobacco), *NtNOX* plays a role in normal pollen tube growth [20]. RBOH-dependent ROS production is also crucial for processes involving cell expansion, such as seed germination, fruit ripening, and the elongation of roots and hypocotyls, where it regulates cell wall loosening [21]. Numerous studies indicate that RBOH proteins are not only involved in growth and development but also mediate responses to various abiotic stresses [22]. For example, *TaRBOHF1* RNAi transgenic wheat exhibits enhanced osmotic regulation and photosynthetic capacity under salt stress, coupled with reduced oxidative damage [12]. In *O. sativa*, knockout of the *OsRBOHE* gene significantly impairs drought resistance compared with wild-type controls [23]. In *Zea mays* (maize), *ZmRBOHB* and *ZmRBOHD* are involved in broad-spectrum responses to abiotic stresses, including temperature extremes, salinity, UV radiation, and drought [24]. However, the functional role of *RBOH* genes in oats has yet to be explored.

Oats (*Avena sativa*), an annual herbaceous plant in the Oat genus of the *Poaceae* family, is well-known for its remarkable tolerance to extreme environmental conditions, including nutrient-poor soils, salinity, and drought. Its broad adaptability makes it a promising candidate for large-scale cultivation [25]. With the rapid expansion of animal husbandry in China, the demand for oats has increased, establishing it as an important forage crop [26,27]. Despite its ecological and economic significance, various factors often limit its yield and quality. Therefore, understanding the genetic traits of oats related to drought and salt tolerance is essential for improving agricultural productivity. This study presents a comprehensive genome-wide analysis of the *RBOH* gene family in oats, including phylogenetic relationships, conserved motifs, chromosomal localization, cis-regulatory elements, and synteny across different species. Additionally, the study investigates the response of *RBOH* genes to various abiotic stressors. The findings offer valuable insights into the role of *RBOH* genes in enhancing oat stress resistance and lay a foundation for future functional studies on these genes.

## 2. Results

### 2.1. Identification and Physicochemical Properties of RBOH Gene Family

A total of 35 non-redundant members of the *AsRBOH* gene family were identified in the oat genome. Analysis of their physicochemical properties revealed that AsRBOH proteins vary in length from 210 to 998 amino acids (Table 1), with molecular weights ranging from 23.82 to 111.34 kDa and an average molecular weight of 96.75 kDa. The isoelectric points (pI) of these proteins ranged from 8.88 to 9.49, suggesting that they have similar charge states near their respective isoelectric points. The instability index of AsRBOH proteins ranged from 34.6 to 55.29, reflecting their stability, with lower values indicating greater protein stability. The Grand Average of Hydropathicity (GRAVY) values ranged from −77.03 to −90.95, indicating the hydrophobicity of the proteins; higher values correspond to more hydrophobic proteins. Additionally, the hydrophilicity index ranged from −0.328 to −0.08, with all values being negative, further suggesting that AsRBOH proteins are predominantly hydrophilic. Overall, AsRBOH proteins are enriched in basic amino acids, and their negative average hydrophilicity coefficient supports their classification as hydrophilic proteins.

### 2.2. Conserved Motifs and Gene Structure Analysis of RBOH Gene Family

The results of the conserved domain analysis are presented in Figure 1A. Members of the *AsRBOH* gene family contain between one and six conserved domains. Specifically, all 30 AsRBOH members possess the NADPH_Ox domain, 28 members include the NOX_Duox_like_FAD_NADP domain, and 20 members harbor the NAD_binding_6 domain. Additionally, a comparison of the genomic and corresponding CDS sequences of *AsRBOH* genes allowed for the determination of their gene structures, as shown in Figure 1B. The majority of *AsRBOH* genes (18 members) consist of two exons, while five members (*5CG0922760*, *4DG0724710*, *1AG0047060*, *1AG0047090*, and *UnG1437940*) contain only one exon. Eleven *AsRBOH* genes are intron-only, and one gene (*3DG0531950*) contains four. Collectively, these findings indicate that the *AsRBOH* gene family in oats is highly conserved.

### 2.3. Chromosomal Localization of RBOH Gene Family

Chromosomal localization analysis revealed that the 35 *AsRBOH* genes are distributed across 14 chromosomes in the oat genome (Figure 2). Further examination revealed an uneven distribution of *AsRBOH* genes among these chromosomes. Specifically, chromosome 1A contains seven *AsRBOH* genes: *1AG0047560*, *1AG0047090*, *1AG0047060*, *1AG0040750*, *1AG0015390*, *1AG0014930*, and *1AG0009010*. Chromosome 1D harbors four *AsRBOH* genes: *1DG0159520*, *1DG0166070*, *1DG0165660*, and *1DG0165630*. Chromosomes 3A, 3C, 3D, 5A, 5C, and 5D each contain three *AsRBOH* genes, while chromosomes 4A, 4C, 4D, 7A, 7D, and the unclassified chromosome (Un) each harbor one *AsRBOH* gene. These findings suggest that the distribution of the *AsRBOH* gene family across the oat genome is not the result of large-scale gene duplication events during genome formation and evolution.

### 2.4. Phylogenetic Analysis of RBOH Gene Family

To explore the evolutionary relationships among the *RBOH* gene family members in oats and other species, this study conducted a multiple sequence alignment of the amino acid sequences from 35 *AsRBOHs* in oats, seven *OsRBOHs* in rice, and 10 *AtRBOHs* in *Arabidopsis* for a total of 52 *RBOHs*. Based on these sequences, a phylogenetic tree was constructed (Figure 3). As illustrated in Figure 3, the *RBOHs* are grouped into four distinct subfamilies, with each subfamily containing members from *Arabidopsis*, rice, and oats. The *AsRBOHs* and *OsRBOHs* within each subgroup show a close evolutionary relationship. Phylogenetic analysis suggests that the *RBOH* gene family predates the divergence of monocots and dicots, with orthologous genes across these species maintaining conserved biological functions.

### 2.5. Synteny Analysis of RBOH Gene Family

To further investigate the evolutionary relationships of the *RBOH* gene family across different species, we performed a comprehensive synteny analysis of the genomes of *A. thaliana*, *Z. mays*, *A. sativa*, *O. sativa*, and *T. aestivum*. The results (Figure 4) revealed significant gene synteny among these five crop species. Specifically, *A. sativa* and *A. thaliana* share five pairs of syntenic genes located on the 1A, 1D, 3C, and 3D chromosomes of *A. sativa*, identified as *1AG0047560*, *1AG0015390*, *3CG0504790*, *1DG0166070*, and *3DG0551470*. A total of 32 pairs of syntenic gene pairs were detected between *A. sativa* and *Z. mays*, distributed across 10 chromosomes: 1A, 1D, 3A, 3C, 3D, 5A, 5C, 5D, 7A, and 7D.

Additionally, 34 pairs of syntenic genes were identified between *A. sativa* and *O. sativa*, with these genes unevenly distributed across 10 chromosomes, including 1A, 1D, 3A, 3C, 3D, 5A, 5C, 5D, 7A, and 7D. In contrast, the synteny relationship between *A. sativa* and *T. aestivum* was closer, with 82 pairs of syntenic genes detected, suggesting a stronger evolutionary connection between *A. sativa* and *T. aestivum* in the *RBOH* gene family, which is more pronounced than the relationships observed with *A. thaliana*, *Z. mays*, and *O. sativa*. Further analysis revealed that the *RBOHs* syntenic genes in *A. sativa* are widely distributed across 10 chromosomes, while no *RBOHs* genes were detected on the remaining 11 chromosomes (2A, 4A, 6A, 1C, 2C, 4C, 6C, 7C, 2D, 4D, 6D). These findings provide important insights into the evolutionary dynamics of the *RBOH* gene family across different species.

### 2.6. Functional Analysis of Cis-Elements in the Promoter Region of RBOH Gene Family

To gain a deeper understanding of the functional roles of the *AsRBOH* gene family, we performed a cis-acting element analysis on the promoter regions (within 2000 bp upstream of the transcription start site) of 35 *AsRBOH* genes (Figure 5). The analysis identified various cis-acting elements associated with abiotic stress, defense responses, plant hormone signaling, and growth and developmental processes. Notably, 24 of the *AsRBOH* gene promoters contained MYB-binding sites linked to drought inducibility, suggesting that *RBOH* genes play a critical role in *A*. *sativa*’s response to environmental stress. Furthermore, 21 enhancer-like elements that are involved in hypoxia-specific induction within the *AsRBOH* promoters were identified. Regarding hormone responsiveness, five distinct types of cis-elements related to plant hormone regulation were detected, including those responsive to abscisic acid, auxin, gibberellin, methyl jasmonate, and salicylic acid. These findings indicate that the *AsRBOH* gene family is not only essential for the growth and development of *A*. *sativa* but also plays a key role in its ability to respond to a variety of environmental stimuli.

### 2.7. Expression Profile of RBOH.s in Response to Abiotic Stress

The expression profiles of 35 *AsRBOH* genes under drought stress were analyzed using quantitative reverse transcription PCR (qRT-PCR), with the results presented in Figure 6A. Under drought conditions, 28 *AsRBOHs* exhibited varying degrees of upregulation. Notably, five genes (*5DG0963470*, *1AG0015390*, *1DG0159520*, *3CG0504790* and *1AG0040750*) reached their peak expression levels 3 h after drought stress. Genes *1AG0047560*, *5AG0850070*, and *1AG0047060* displayed the highest expression levels at 6 h post-stress. Fourteen *AsRBOHs* (*4AG0580180*, *4CG1304250*, *5DG0988170*, *5DG0963470*, *5CG0887860*, *3DG0550720*, *1AG0014930*, *5DG0959800*, *3DG0551470*, *5AG0850070*, *1AG0047060*, *3DG0531950*, *3CG0479430*, and *3AG0430660*) peaked at 12 h post-stress, while six genes (*7DG1382190*, *7AG1225850*, *3AG0449960*, *5CG0916260*, *5AG0846340*, and *UnG1437940*) showed highest expression level at 24 h post-drought stress. Conversely, seven *AsRBOHs* genes (*1DG0166070*, *1DG0165660*, *1AG0047090*, *1DG0165630*, *1AG0009010*, and *5CG0922760*) exhibited significant downregulation under drought stress. These findings suggest that *AsRBOHs* are involved in oat responses to drought stress through complex and time-dependent regulatory mechanisms.

Similarly, the expression of the 35 *AsRBOH* genes was assessed under salt stress. Compared with the control group (0 h), 17 *AsRBOHs* (*5DG0963470*, *5DG0988170*, *UnG1437940*, *1AG0009010*, *4DG0724710*, *4AG0580180*, *1AG0015390*, *1AG0047560*, *5CG0916260*, *5AG0820170*, *1DG0166070*, *1DG0165630*, *1AG0047090*, *1DG0165660*, *1AG0047060*, *3AG0449960*, and *1AG0014930*) were downregulated, while the remaining 18 *AsRBOHs* displayed varying degrees of upregulation. Notably, *5CG0922760* and *3DG0550720* showed the highest expression levels at 3 h post-salt stress, whereas *4CG1304250* and *5CG0887860* peaked at 6 h. Thirteen genes (*7DG1382190*, *7AG1225850*, *3CG0504180*, *5AG0850070*, *3CG0479430*, *5DG0959800*, *1DG0159520*, *1AG0040750*, *3DG0531950*, *3AG0430660*, *3CG0504790*, *3AG0450850*, and *3DG0551470*) also showed significant expression changes under salt stress and reached peak expression at 12 h. Only *5AG0846340* reached its peak expression at 24 h post-salt stress with moderate expression level (Figure 6B). These results demonstrate significant variability in both the timing and magnitude of *AsRBOH* gene responses to salt stress, indicating differential regulatory patterns among the genes.

### 2.8. Spatiotemporal Expression Characteristics of RBOH Gene Family in Oat

This study systematically analyzed the spatiotemporal expression characteristics of 35 *AsRBOHs* genes during different seed developmental stages using oat public expression data (PRJEB46365). Results showed that the majority of *AsRBOHs* exhibited their highest relative expression levels during the early developmental stage (including both daytime and nighttime samples). Two genes (*1AG0014930* and *1AG0047090*) maintained low expression levels throughout all developmental phases. Four genes (*5DG0988170*, *5CG0916260*, *3DG0551470*, and *UnG1437940*) reached peak expression during the mid-developmental stage without showing diurnal variation. Notably, distinctive day-night expression divergence emerged in the late developmental stage: one genes *1AG0015390* specifically peaked during daytime samples, while eleven genes (*4AG0580180*, *7DG1382190*, *1DG0166070*, *4DG0724710*, *1AG0047060*, *1DG0165630*, *1DG0165660*, *5CG0922760*, *5AG0820170*, *7AG1225850*, and *3CG0504180*) showed expression maxima in nighttime samples (Figure 7).

## 3. Discussion

The *RBOH* gene family plays a critical role in plant development and stress responses [28]. It has been widely observed across various plant species, with most members being highly conserved [29,30]. Research on the *RBOH* gene family has predominantly focused on dicotyledonous plants, such as *A. thaliana*, *T. aestivum*, and *Z. mays* [4]. Oat (*A. sativa*), a key annual grass species in the *Poaceae* family, holds significant economic and ecological value [31]. In this study, we identified 35 *AsRBOH* genes in the oat genome. Interestingly, while the number of *RBOH* gene family members varies across different plant species, it does not directly correlate with genome size [32]. The rice genome has a size of 385.7 Mb, with nine members of the *RBOH* gene family identified. In contrast, the quinoa genome is 1.54 Gb in size and also contains nine members of the *RBOH* gene family [33]. Phylogenetic analysis of *AsRBOHs*, compared to the *AtRBOHs* in *Arabidopsis* and the *OsRBOHs* in rice, revealed that the 35 *AsRBOHs* can be classified into five distinct subfamilies. Notably, most *AsRBOH* proteins cluster closely with *OsRBOHs*, suggesting a closer evolutionary relationship. Gene synteny analysis further indicated that *AsRBOHs* share the highest number of syntenic gene pairs with *RBOH* genes from *T. aestivum* and *O. sativa*, reinforcing the conserved nature of *RBOH* genes within the *Poaceae* family. This observation underscores the evolutionary similarities between oats and wheat, as well as the broader conservation of the *RBOH* family across diverse plant species [34,35].

Cis-acting elements, specific DNA sequences located near structural genes, serve as binding sites for transcription factors. Transcription factors regulate gene expression at the transcriptional level by interacting with these elements, thereby modulating plant responses to abiotic stress and influencing growth and development [36,37]. In *A. thaliana*, the promoters of 10 *RBOH* genes contain various stress-responsive elements, including motifs associated with salt, cold and drought stress, as well as hormone-responsive and growth-related cis-acting elements [13]. In this study, several abiotic stress response elements were identified within the promoters of *AsRBOH* genes. Notably, the promoter sequences of 24 *AsRBOH* genes contained MYB-binding site motifs associated with drought inducibility, suggesting a crucial role for *RBOHs* in oat stress response. Additionally, 21 elements resembling enhancers involved in hypoxia-specific induction were detected in the promoter regions of *AsRBOHs*. These findings align with similar observations in *Arabidopsis* [13] and *C. quinoa* [16], indicating that the *AsRBOH* family members are likely involved in regulating responses to abiotic stress. These cis-acting elements enable plants to rapidly adapt to environmental challenges, thereby enhancing their resilience against stresses.

Plants exposed to abiotic stresses, such as drought and high salinity, rapidly accumulate reactive oxygen species (ROS), leading to oxidative damage [38]. NADPH oxidases, specifically the respiratory burst oxidase homologs (*RBOHs*), are pivotal in ROS production and hydrogen peroxide (H_2_O_2_) accumulation, processes regulated by *RBOH* genes [39]. Numerous studies have linked *RBOH* family members to abiotic stress responses. For instance, silencing the *SiRboh1* gene in tomatoes reduces drought tolerance [40], and *RBOH* genes in soybeans (*Glycine max*) are significantly induced under salt and polyethylene glycol (PEG) stress [41]. In chili peppers, the expression of *CaRbohB* and *CaRbohD* significantly increases under salt and drought treatments [42]. In this study, expression pattern analysis of the *AsRBOH* gene family under salt and drought conditions revealed that most *AsRBOH* genes are involved in abiotic stress responses. Under drought stress, the expression of 28 *AsRBOH* genes was upregulated, with 14 reaching peak expression 12 h after exposure, suggesting a positive regulatory role in drought adaptation. These genes are promising candidates for further research into drought stress mechanisms in oats. Previous research has demonstrated that overexpression of *TaSNAC11*-*4B* in *Arabidopsis* enhances drought-induced senescence by binding to the promoters of *AtRbohD*/*F*, increasing ROS accumulation, and promoting programmed cell death to help the plant adapt to drought conditions [12]. Under salt stress, 18 *AsRBOH* genes were upregulated, indicating a positive regulatory role in salt stress response. Conversely, 17 *AsRBOH* genes were downregulated under salt stress, suggesting inhibition during salt exposure. Studies have shown that knocking out *NtRbohE* in *NtbHLH123*-overexpressing plants reduces the expression of ROS scavenging and salt stress-related genes, resulting in a salt tolerance decrease [43]. In pigeon peas (*Cajanus cajan*), *CcRbohG* and *CcRbohH* are significantly upregulated under both drought and salt stress [44]. Thus, *AsRBOH* genes may serve as critical targets for enhancing salt stress tolerance in oats.

Interestingly, genes *1DG0165660* and *1AG0047090*, which belong to the same subfamily, exhibited similar expression patterns under both drought and salt treatments, suggesting functional redundancy to some extent. However, functional differentiation within the same subfamily was evident, as demonstrated by the significant upregulation of *3AG0449960* under drought stress, while *3CG0504180* was strongly repressed. Additionally, both *7DG1382190* and *7AG1225850* showed significant upregulation in response to both salt and drought stress, indicating their critical roles in oats’ abiotic stress responses.

These findings suggest that the *AsRBOH* genes regulate the accumulation of reactive oxygen species (ROS) in oats, thereby contributing to cell signaling and senescence processes triggered by salt and drought stress. The diverse and complex roles of these genes in mediating abiotic stress responses warrant further investigation to unravel their full potential in improving stress tolerance in oats.

## 4. Materials and Methods

### 4.1. Plant Materials and Methods

Oats (*Avena sativa*) seeds of the cultivar Zhongxuyan No.1 were selected for germination in Petri dishes, then transplanted into seedling trays and grown in a greenhouse with a relative humidity of 65–75%. When the seedlings reached the three-leaf stage, drought and salt stress treatments were applied. Drought stress was simulated using 20% PEG 6000 (Mei5bio, Beijing, China), and samples were collected at five time points (0 h, 3 h, 6 h, 12 h, and 24 h). Salt stress was induced using 150 mM NaCl (Mei5bio, Beijing, China), with samples collected at the same time points. After collection, samples were immediately frozen in liquid nitrogen and stored at −80 °C for subsequent RNA extraction. Each treatment included three biological replicates.

### 4.2. Identification Analysis of RBOH Gene Family

Ten AtRBOH protein sequences were downloaded from the *Arabidopsis* Genome Database (TAIR, https://www.arabidopsis.org/, accessed on 10 April 2024) and used as query sequences in a BLASTP search against oat *RBOHs* (E-value < 1 × 10^−5^) using TBtools v1.121. The complete oat genome data (GCA_022788535) was downloaded from the Ensembl Plants database (https://plants.ensembl.org/, accessed on 6 May 2024), and the oat *RBOHs* were identified using the Hmm search program in TBtools v1.121. The protein sequences obtained from both methods were merged, and redundant sequences were removed. The final oat *RBOH* gene family members were identified. Candidate proteins were further verified using Pfam (http://pfam.xfam.org/, accessed on 26 May 2024) and NCBI-CDD (https://www.ncbi.nlm.nih.gov/cdd, accessed on 2 July 2024). The molecular weight and isoelectric points of the oat RBOHs protein sequences were predicted using TBtools v1.121.

### 4.3. Conserved Motifs and Gene Structure Analysis

The gene structure of the oat *RBOH* family members was analyzed using TBtools v1.121. The conserved motifs in the oat RBOHs proteins were analyzed using MEME (http://meme.nbcr.net/meme, accessed on 13 July 2024), with the maximum number of motifs set to 10. Visualization of the results was performed using TBtools v1.121.

### 4.4. Chromosomal Localization Analysis of RBOH Gene Family

The oat gene density file was extracted using the Chromosome Heatmap Data function of TBtools v1.121. The chromosomal positions of the *AsRBOH* gene family members were determined based on the oat genome database. The distribution of *AsRBOHs* genes on the chromosomes was visualized using the Gene Location Visualize from GTF/GFF tool in TBtools v1.121.

### 4.5. Evolutionary Analysis of the AsRBOH System

The AtRBOH protein sequences were downloaded from TAIR (https://www.arabidopsis.org/, accessed on 28 July 2024), and the RBOH protein sequences from rice were obtained from NCBI (https://www.ncbi.nlm.nih.gov/, accessed on 8 August 2024). A multiple sequence alignment of the 52 RBOH protein sequences from oat, rice, and Arabidopsis was performed using MUSCLE in MEGA-X. A phylogenetic tree was constructed using the Neighbor-Joining (NJ) method in MEGA v7.0 software, and the tree was visualized and refined in iTOL (https://itol.embl.de, accessed on 19 August 2024).

### 4.6. Synteny and Homologous Gene Pairs

Synteny analysis between the *RBOH* gene families of *A. thaliana*, *Z. mays*, *O. sativa*, *T. aestivum*, and *A. sativa* was conducted using the One Step MCScanX module in TBtools. The synteny chromosome distribution of *AsRBOHs* genes was visualized using the Dual Systeny Plot plugin in TBtools, which illustrates the genomic locations of *AsRBOHs* genes, their syntenic relationships, and their correspondence with *RBOHs* genes from other species.

### 4.7. Promoter Analysis of the RBOH Gene Family

Based on the reference genome assembly of cultivated oat (*A. sativa*, accession number GCA_022788535), we systematically retrieved the 2000 bp upstream promoter regions of all identified *RBOH* (Respiratory Burst Oxidase Homolog) gene family members. Subsequently, cis-acting regulatory elements within these promoter sequences were comprehensively annotated using the Plant CARE database (Plant Cis-acting Regulatory Element Database; http://bioinformatics.psb.ugent.be/webtools/plantcare/html/, accessed on 7 October 2024). The spatial distribution and functional classification of the predicted cis-elements were further visualized through the TBtools bioinformatics platform.

### 4.8. Expression Pattern of RBOH Gene in Oat

Total RNA from oat plants subjected to drought and salt stress treatments was extracted using the Fast Pure Plant Total RNA Extraction Kit (Zomanbio, Beijing, China). cDNA synthesis was carried out using the Trans Script II One-Step gDNA Removal and cDNA Synthesis Super Mix Kit (Vazyme, Nanjing, China). qRT-PCR was performed using a Light Cycle 96 with Hieff® qPCR SYBR Green Master Mix Kit (Yeasen, Shanghai, China), with a reaction system of 10 μL containing 5.0 μL SYBR Green Master Mix, 0.4 μL of each upstream and downstream primer, 3.2 μL sterile water, and 1 μL cDNA template.

The PCR program was as follows: 95 °C for 3 min for pre-denaturation, followed by 15 s at 95 °C for denaturation, 30 s at 60 °C for annealing, and 20 s at 72 °C for extension, with 40 cycles. A melting curve analysis was conducted from 60 to 95 °C. AsActin was used as the internal reference gene, and relative gene expression levels were calculated using the 2^-ΔΔCt^ method. All experiments were conducted with three biological replicates. The qRT-PCR primer sequences are listed in Appendix A. The RNA-seq data were obtained from the NCBI website (https://www.ncbi.nlm.nih.gov/, accessed on 27 October 2024), utilizing public oat expression data (PRJEB46365) across different developmental stages of oat seeds.

### 4.9. Data Analysis

The qRT-PCR data were organized using Excel 2019, and a heatmap representing the relative expression levels of the 35 *AsRBOHs* genes under different abiotic stress conditions was generated using GraphPad Prism v10.0 software. Cluster analysis of the expression data was performed to evaluate the patterns further.

## 5. Conclusions

In this study, 35 *AsRBOH* genes were identified in the oat genome, distributed across 14 chromosomes. Phylogenetic analysis grouped these genes into five subfamilies, with genes within the same subfamily sharing similar structures and motifs. The promoter regions of the *AsRBOH* genes contain numerous cis-acting elements linked to responses to abiotic stresses, including drought and other environmental stressors. These genes play a key role in regulating oat responses to drought and salt stress, with *7DG1382190* and *7AG1225850* identified as pivotal genes involved in these stress responses. These findings offer valuable insights into the function and regulatory mechanisms of the *AsRBOH* gene family in oats and provide a foundation for further investigation into their roles in oat physiology and stress tolerance regulation mechanisms.

## Figures and Tables

**Figure 1 ijms-26-02576-f001:**
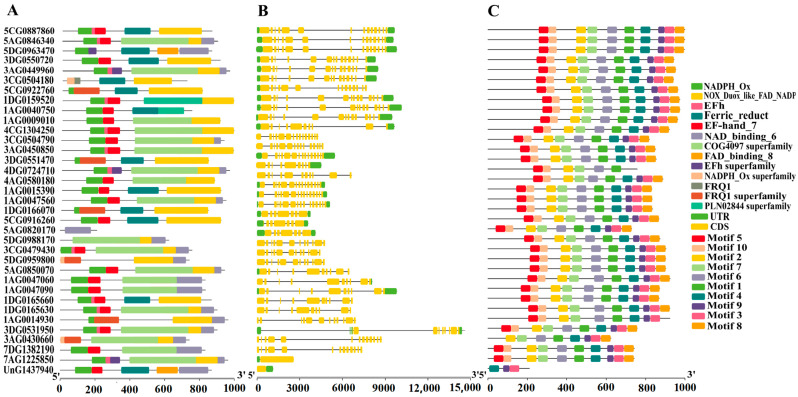
(**A**) AsRBOHs protein conserved structural domains; (**B**) Gene structure analysis of the *RBOH* gene family. (**C**) Conserved motifs analysis of the *RBOH* gene family.

**Figure 2 ijms-26-02576-f002:**
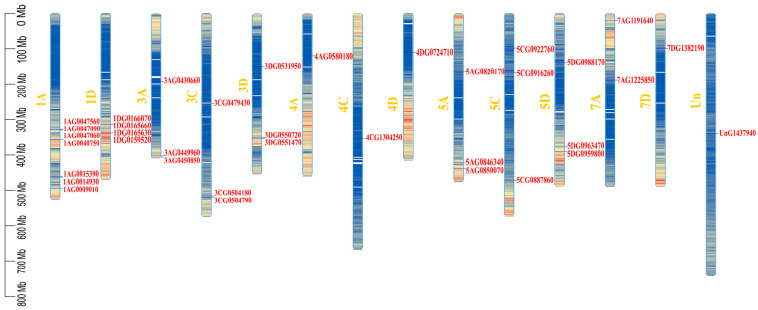
Chromosomal locations of the *RBOH* gene family. Note: Gene density is depicted in a color gradient, with red indicating regions of higher gene density and blue indicating regions of lower gene density.

**Figure 3 ijms-26-02576-f003:**
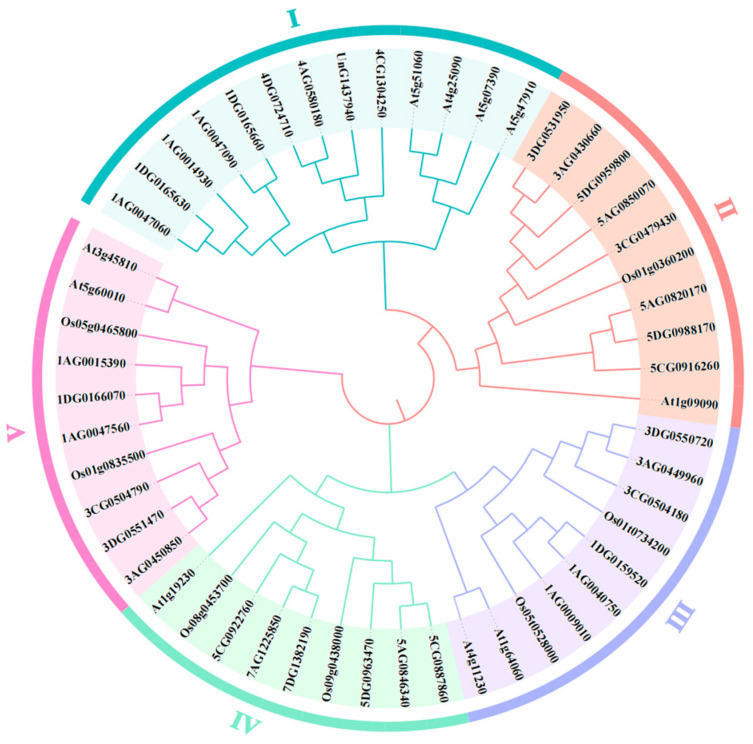
Phylogenetic tree of *RBOH* gene family. Note: Clades I–IV are color-coded to indicate the evolutionary groupings of the RBOH proteins. The gene accession numbers for the rice *RBOHs* family *OsRBOHsA*~*G* are *Os01t0734200*, *Os01g0360200*, *Os05t0528000*, *Os05g0465800*, *Os01g0835500*, *Os08g0453700*, and *Os09g0438000*. The gene accession numbers for the *Arabidopsis RBOHs* family *AtRBOHsA*-*J* are *At5g07390*, *At5g51060*, *At5g47910*, *At1g19230*, *At1g64060*, *At4g25090*, *At5g60010*, *At4g11230*, and *At3g45810*.

**Figure 4 ijms-26-02576-f004:**
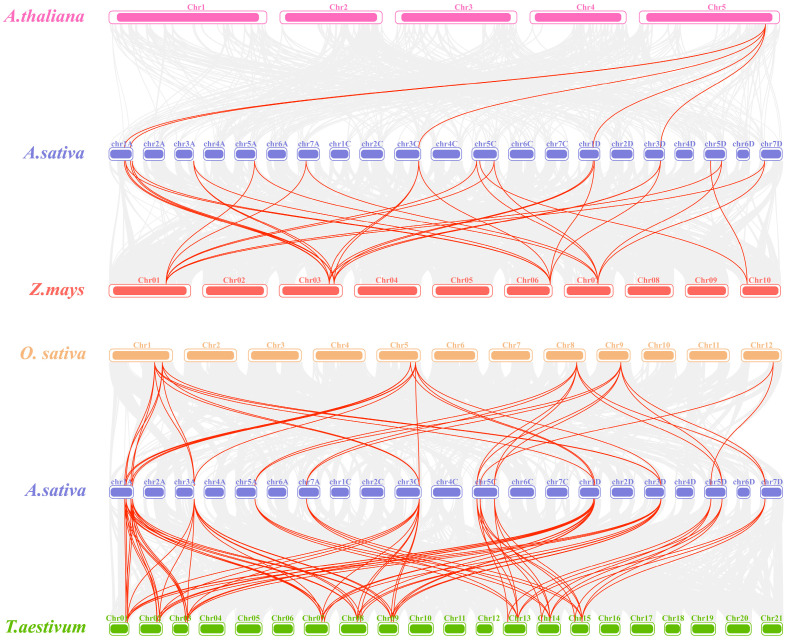
Presents an analysis of *RBOH* gene collinearity between *A*. *sativa* and four representative plant species. The gray lines in the background highlight collinear blocks between *A. sativa* and the other species (*A. thaliana*, *Z. mays*, *O. sativa* and *T. aestivum*), while the red lines indicate pairs of homologous *RBOH* genes. The grey lines in the background indicate blocks of collinearity within *A. sativa* and the indicated plants, whereas the red lines indicate homozygous *RBOH* gene pairs.

**Figure 5 ijms-26-02576-f005:**
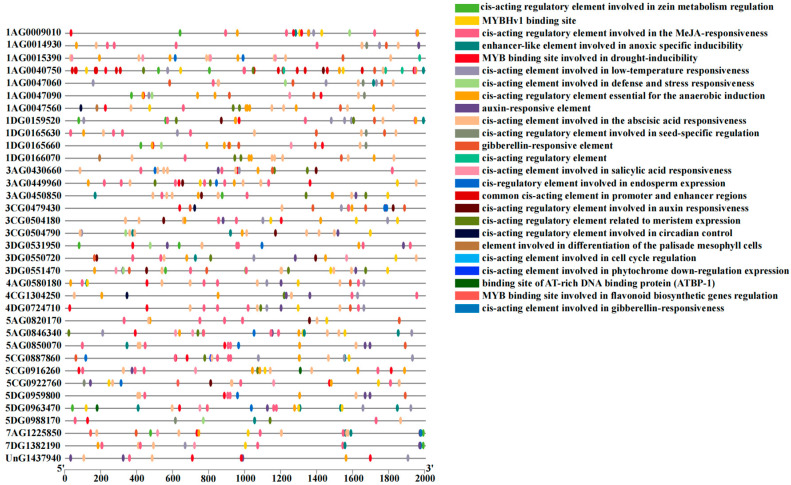
Distribution of cis-acting elements in *RBOH* gene family promoters. Note: Genomic segments were standardized to 2000 bp upstream of transcription start sites (TSS) for comparative analysis.

**Figure 6 ijms-26-02576-f006:**
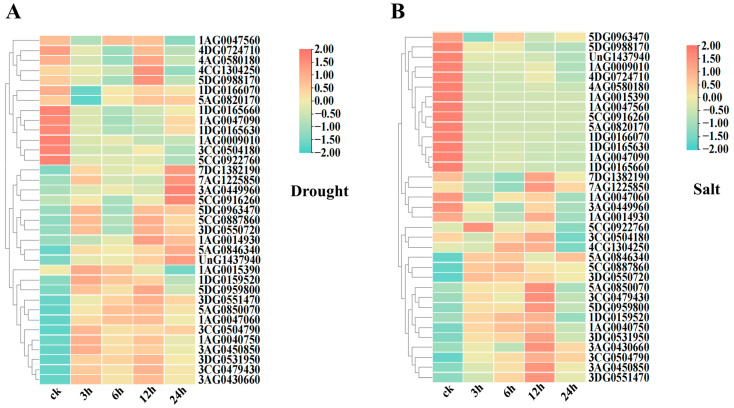
(**A**) Relative expression of *AsRBOH* gene under PEG stress. (**B**) Relative expression of *AsRBOH* gene under NaCl stress.ck: non-stressed control plants harvested at matching time points.

**Figure 7 ijms-26-02576-f007:**
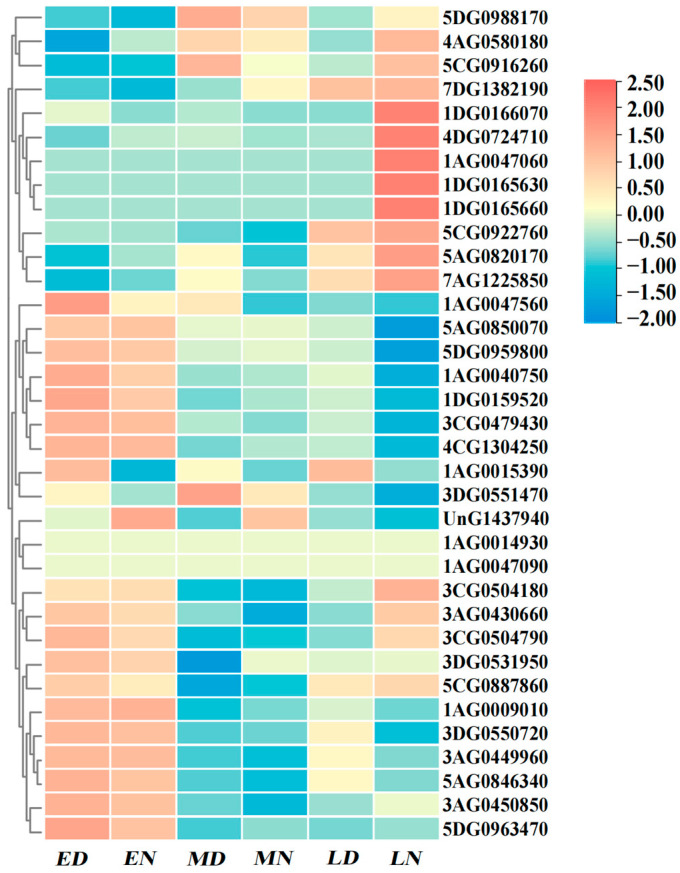
Expression patterns of *AsRBOH* genes at different developmental stages of oat seeds. Note: ED: Oat RNA-seq seed early development stage day. EN: Oat RNA-seq seed early development stage night. MD: Oat RNA-seq seed middle development stage day. MN: Oat RNA-seq seed middle development stage night. LD: Oat RNA-seq seed late development stage day. LN: Oat RNA-seq seed late development stage night.

**Table 1 ijms-26-02576-t001:** Physical and chemical properties of RBOH gene family.

ID	Number of Amino Acid	Molecular Weight	Theoretical pI	Instability Index	Aliphatic Index	Grand Average of Hydropathicity
5CG0887860	998	111,341.98	9.15	44.25	81.39	−0.201
5AG0846340	998	111,338.95	9.17	45.65	81.51	−0.21
5DG0963470	996	110,913.53	9.24	45.58	81.97	−0.188
3DG0550720	944	107,073.76	9.23	50.33	85.81	−0.275
3AG0449960	953	107,773.47	9.2	49.74	85.31	−0.265
3CG0504180	942	106,712.39	9.23	51.15	85.77	−0.271
5CG0922760	964	108,040.8	9.37	55.29	83.31	−0.154
1DG0159520	973	108,865.93	9.26	50.09	86.09	−0.213
1AG0040750	974	108,991.03	9.34	50.13	86.3	−0.218
1AG0009010	963	108,188.18	9.34	50.27	86.77	−0.228
4CG1304250	920	103,729.73	9.35	45.25	81.84	−0.275
3CG0504790	818	92,699.45	9.49	42.94	86.44	−0.151
3AG0450850	850	95,816.81	9.43	43.96	84.8	−0.172
3DG0551470	853	96,036.01	9.43	44.38	84.16	−0.174
4DG0724710	757	84,801.24	9.37	50.46	83.75	−0.224
4AG0580180	888	100,001.28	9.33	46.29	82.82	−0.283
1AG0015390	832	94,172.33	8.98	48.11	80.66	−0.219
1AG0047560	833	94,049.25	9.1	46.69	82.55	−0.198
1DG0166070	834	94,189.35	9.05	46.35	82.22	−0.206
5CG0916260	868	97,724.17	9.1	43.25	86.14	−0.193
5AG0820170	729	82,589.16	8.88	41.6	87.17	−0.089
5DG0988170	872	97,993.43	8.97	44.97	85.4	−0.181
3CG0479430	903	101,712.85	9.28	39.21	85.3	−0.203
5DG0959800	902	101,652.54	9.21	35.66	84.63	−0.225
5AG0850070	902	101,603.6	9.24	36.3	85.28	−0.211
1AG0047060	924	103,793.61	9.02	40.78	79.59	−0.28
1AG0047090	871	98,458.01	9.4	42.87	81.93	−0.274
1DG0165660	869	98,101.61	9.35	45.26	81.24	−0.266
1DG0165630	923	104,207.28	9.14	43.22	77.03	−0.325
1AG0014930	923	103,515.24	9.22	47.1	78.86	−0.328
3DG0531950	757	86,159.29	8.98	34.6	85.01	−0.14
3AG0430660	624	71,049.03	9.19	37.12	86.71	−0.083
7DG1382190	741	84,588.75	9.38	50.19	87.04	−0.087
7AG1225850	741	84,470.55	9.3	48.52	87.3	−0.08
UnG1437940	210	23,824.27	9.28	35.79	90.95	−0.26

## Data Availability

The oat data provided in the study have been deposited in the Ensembl Plants web repository at (https://plants.ensembl.org/Avena_sativa_Sang/Info/Index, accessed on 30 October 2024), while rice protein information is accessible through (https://plants.ensembl.org/Oryza_sativa/Info/Index, accessed on 2 November 2024). Wheat protein information, with the access number GCF_018294505.1, can be found on the NCBI website (https://www.ncbi.nlm.nih.gov, accessed on 10 November 2024). The protein sequences corresponding to the *RBOH* transcription factor subfamily genes in *Arabidopsis* were retrieved from The *Arabidopsis* Information Resource (https://www.arabidopsis.org/, accessed on 15 November 2024). The datasets supporting the conclusions of this study are included in the article and in additional files.

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
