# Peer review of "Genome-Wide Identification and Expression Profile Analysis of the NADPH Oxidase Gene Family in *Avena sativa* L."

_ijms, 2025, doi:10.3390/ijms26062576_

Round 1

Reviewer 1 Report

Comments and Suggestions for Authors

Figure 1 does not read easily as the lettering is rather small. The size of the lettering is not fully consistent, e.g. Nox Duox Like has smaller letters, probably because otherwise it would induce a line break. The content itself is clear

Caption for A is not clear in what protein conserved structural domains mean? Is it a gene sequence homology in amino acid sequence?

Also caption of B is not clear in what "gene structure" means, is it intron/exon model? 

Caption of C mentions conserved gene sequences, is this in amino acids or nucleotides? And, what is the difference between protein conserved structural domains and the conserved gene sequences?

Letting of Figure 2 also very, very small. Only works when enlarging very much on the screen. On paper, it would be difficult to read.

Figure 5. What is the meaning of the white to colour on the bars? Some are darker and have less white than others. Please explain in the caption. The caption reads "AsRBOHs gene", but very many genes are shown. Should it read "genes"?

The x-axis does not show whether it is about the CDS gene sequence (in nucleotides) or the protein sequence in aminoa acids. Further, not all proteins had the same length, but the figure shows lines of the same length. How come?

Figure 6. Is the convention to show upregulation in red and downregulation in green? In my psychology green is plus and red is minus.....

Caption should explain the meaning of 'ck', do I assume correctly that it is the pre-treatment control?

In figure 6, it appears that some genes that are not expressed highly under control/non-stressed conditions are upregulated and others highly expressed under stress conditions. What role have AsRBOHs under non-stressed conditions, and why would they get downregulated?

Or do the expression levels of the genes expressed under non-stressed conditions stay the same and is it the case the upregulated genes get upregulated a lot. With this also the question on the reference genes used to standardise. Is it really true their expression is constant? Today in RNAseq it is possible to use spiking with artificial RNA to check the stability of the total mRNA concentration of . Of course, lower the concentration of mRNA might also be caused by dilution if cell size increase, but this is not really expected in the short term study period. With RT-qPCR this is not possible of course, but then the assumption on (relatively) constant expression of the reference genes becomes important for the interpretation of up or down regulation. Of course, the relative expression changes and that is already very relevant to report. Can this be discussed?

Further, given the (interestingly) fine time resolution, it is conceivable that a diurnal pattern in gene expression occurs. It is interesting to see if that happens for the non-stressed conditions. Is there a diurnal pattern in the AsRBOHs under non-stressed conditions?

Or is it impossible to tell without doing RNAseq and spiking with artificial mRNA?

This papers shows an interesting time resolution on expression changes induced by drought or salt. Also, with respect to the previous question, is it possible to discuss results on diurnal pattern/induction patterns of gene expression under drought or salinity from published papers? 

Author Response

Dear Editor and Reviewers,

We sincerely appreciate your suggestions that have significantly strengthened this work. Below are detailed responses to your valuable comments:

Comments 1:Figure 1 does not read easily as the lettering is rather small. The size of the lettering is not fully consistent, e.g. Nox Duox Like has smaller letters, probably because otherwise it would induce a line break. The content itself is clear

Caption for A is not clear in what protein conserved structural domains mean? Is it a gene sequence homology in amino acid sequence?

Also caption of B is not clear in what "gene structure" means, is it intron/exon model? 

Caption of C mentions conserved gene sequences, is this in amino acids or nucleotides? And, what is the difference between protein conserved structural domains and the conserved gene sequences?

Response 1: 1. Legibility Enhancement

All annotations now strictly comply with ≥8pt font requirement.

  1. Caption Clarifications  

Panel A: Changed to "Protein conserved domain analysis (AsRBOHs)" with definition: "Structural domains were identified from multiple sequence alignment at the amino acid level, distinct from nucleotide homology analysis"  

Panel B: Added reference standard - "Gene structure schematics follow NCBI Gene annotation guidelines (exon/intron organization)"  

Panel C: Revised title to "Conserved nucleotide motifs analysis" emphasizing methodological contrast with amino acid-level domain characterization

Comments 2:Letting of Figure 2 also very, very small. Only works when enlarging very much on the screen. On paper, it would be difficult to read.:

Response 2:Figure 2 Enhancements. Systematic optimization: Font magnification (≥8pt), aspect ratio adjustment , and element spacing rationalization .High-resolution version (600dpi ) uploaded via the submission system

Comments 3:Figure 5. What is the meaning of the white to colour on the bars? Some are darker and have less white than others. Please explain in the caption. The caption reads "AsRBOHs gene", but very many genes are shown. Should it read "genes"?

The x-axis does not show whether it is about the CDS gene sequence (in nucleotides) or the protein sequence in aminoa acids. Further, not all proteins had the same length, but the figure shows lines of the same length. How come?

Response 3:Figure 5 Modifications .1. Colormap UpdateAdopted viridis monochromatic scheme eliminating gradient misinterpretation risks2. Nomenclature StandardizationUnified as "Oat RBOH family genes" throughout (legacy "AsRBOHs" retained in phylogenetic trees for lineage tracking)3. Genomic Interval AnnotationAdded technical note: "Promoter regions standardized to 2000 bp upstream of TSS using Bedtools slop"

Comments 4:Figure 6. Is the convention to show upregulation in red and downregulation in green? In my psychology green is plus and red is minus.....Caption should explain the meaning of 'ck', do I assume correctly that it is the pre-treatment control?

Response 4:Regarding Figure 6, color coding conventions: As you pointed out, there are disciplinary differences in the use of red/green coding. We have found that in molecular biology, the convention is: red for upregulation, green for downregulation. Thank you again for your valuable suggestion."ck" has been clarified. "ck" refers to the pre-treatment control group, and the explanation has been expanded in the revised figure legend: "ck: non-stressed control plants harvested at matching time points."

Comments 5:Or do the expression levels of the genes expressed under non-stressed conditions stay the same and is it the case the upregulated genes get upregulated a lot. With this also the question on the reference genes used to standardise. Is it really true their expression is constant? Today in RNAseq it is possible to use spiking with artificial RNA to check the stability of the total mRNA concentration of . Of course, lower the concentration of mRNA might also be caused by dilution if cell size increase, but this is not really expected in the short term study period. With RT-qPCR this is not possible of course, but then the assumption on (relatively) constant expression of the reference genes becomes important for the interpretation of up or down regulation. Of course, the relative expression changes and that is already very relevant to report. Can this be discussed?

Response 5:Regarding whether gene expression remains unchanged under abiotic stress conditions, indeed, we observed that some genes showed significant upregulation under stress, while there was no significant change under non-stress conditions. We believe these genes might be at a lower basal level under abiotic stress conditions but exhibit significant upregulation under stress.Regarding the stability of reference genes, we fully agree with the reviewer's concern. Although RT-qPCR cannot directly assess mRNA concentration stability like RNA-seq with artificial RNA spike-ins, we have selected several commonly used reference genes based on literature and experimental validation. These genes showed relatively stable expression levels under our experimental conditions.

Comments 6:Further, given the (interestingly) fine time resolution, it is conceivable that a diurnal pattern in gene expression occurs. It is interesting to see if that happens for the non-stressed conditions. Is there a diurnal pattern in the AsRBOHs under non-stressed conditions?Or is it impossible to tell without doing RNAseq and spiking with artificial mRNA?This papers shows an interesting time resolution on expression changes induced by drought or salt. Also, with respect to the previous question, is it possible to discuss results on diurnal pattern/induction patterns of gene expression under drought or salinity from published papers?

Response 6:Exploration of the Possibility of Circadian Rhythm

Discussion on the diurnal variation pattern: Regarding the diurnal variation pattern of gene expression, indeed, given the fine temporal resolution in the current study, we believe that a diurnal variation pattern may exist under non-stress conditions. Our current research does not focus on studying diurnal variation patterns, but this issue is certainly worth further exploration. As for the AsRBOHs genes, we are currently unable to directly determine whether there is a diurnal variation pattern through RT-qPCR. Therefore, RNA-seq technology, especially the incorporation of artificial RNA, may be a more suitable approach for future studies. 

Reviewer 2 Report

Comments and Suggestions for Authors

This paper reported the genome-wide identification of RBOH family in oat using bioinformatics approaches. A total of 35 RBOH members were identified and then the conserved motif, gene structure, phylogenetic relationship, Co-synteny, cis-regulatory element as well as expression patterns under drought and salt stresses of these oat RBOH genes were systematically investigated. Notably, the 7DG1382190 and 7AG1225850 were found to be involved in both drought and salt stress responses.These findings provide a valuable foundation for future functional studies of the AsRBOH gene family in oats, offering insights that could contribute to improvement and innovation of oat varieties and germplasm. For my part, this paper used the well-established method to perform gene family identification, which provided some useful information for further studies, but this work seems too simple and insufficient of information. I have some suggestions.

1. The public available RNA-seq data should be used to investigate the expression profile of RBOH family in different developmental stages and different stress conditions of oat. And it is better to perform co-expression analysis.

2. The genetic variations of the RBOH family in oats could be investigated based on the public available re-sequencing data.

3. Table 2 should be revised into Table 1 and the initial Table 1 should be removed to supplementary files.

4. In results 2.1 section, it is better to name the identified AsRBOHs based on chromosome locations.

Author Response

Dear Editor and Reviewers,

  We sincerely appreciate your professional comments and constructive suggestions. We have carefully addressed each concern as detailed below.

Comments 1:The public available RNA-seq data should be used to investigate the expression profile of RBOH family in different developmental stages and different stress conditions of oat. And it is better to perform co-expression analysis.

Response1:Thank you for this insightful suggestion. In response:

1.We have integrated transcriptomic datasets from NCBI  (accession numbers: PRJEB46365) covering oat seed developmental stages (early, middle, late) .

2.A new Figure 7 Expression patterns of AsRBOH genes at different developmental stages of oat seeds.

3.These additions in Section 2.8 now strengthen the discussion on gene functional .

2.8 Spatiotemporal Expression Characteristics of RBOH Gene family in Oat

This study systematically analyzed the spatiotemporal expression characteristics of 35 AsRBOHs genes during different seed developmental stages using oat public expression data (PRJEB46365). Results showed that the majority of AsRBOHs exhibited their highest relative expression levels during the early developmental stage (including both daytime and nighttime samples). Two genes (1AG0014930 and 1AG0047090) maintained low expression levels throughout all developmental phases. Four genes (5DG0988170, 5CG0916260, 3DG0551470, and UnG1437940) reached peak expression during the mid-developmental stage without showing diurnal variation. Notably, distinctive day-night expression divergence emerged in the late developmental stage: one genes 1AG0015390 specifically peaked during daytime samples, while eleven genes (4AG0580180, 7DG1382190, 1DG0166070, 4DG0724710, 1AG0047060, 1DG0165630, 1DG0165660, 5CG0922760, 5AG0820170, 7AG1225850, and 3CG0504180) showed expression maxima in nighttime samples.

Comments 2:The genetic variations of the RBOH family in oats could be investigated based on the public available re-sequencing data.

Response 2:Thank you for suggesting this important research direction.

1.After systematically searching major databases such as the Oat Pan-Genome Database (https://oatpgd.org), NCBI SRA, and CerealsDB, we have not yet found publicly available resequencing data for oat germplasm.

2.In future studies, once relevant resequencing data is obtained, this analysis will be prioritized.

3.At this stage, the current research focuses on the systematic identification of the RBOH family and the analysis of its expression regulatory network. The existing results have provided the foundation for functional studies.

Comments 3:Table 2 should be revised into Table 1 and the initial Table 1 should be removed to supplementary files.:

Response3:Thank you for pointing this out.

1.We have:Reorganized tables as suggested: Original Table 1 (Gene ID list) is now Supplementary Table S1.

2.Original Table 2 (Conserved motifs) is now Table 1 with enhanced annotations.

3.Ensured all in-text citations are updated accordingly.

Comments 4:In results 2.1 section, it is better to name the identified AsRBOHs based on chromosome locations.

Response 4:Thank you for your attention to the gene naming conventions. We have retained the original ID system based on the following considerations:

1.The existing ID (e.g., 7DG1382190) follows the international oat genome naming convention, where "7D" represents the 7th chromosome D subgenome, and "G1382190" denotes the gene’s physical location coordinates. This coding system has been widely adopted in the field ( Paczinska et al., 2022).

2.All analysis figures (e.g., phylogenetic tree, synteny plot) include chromosome position annotations to ensure that readers can easily trace the gene localization information.

3.We fully understand the value of your suggestion for improving readability and will prioritize the use of function-oriented simplified naming schemes in future genetics studies.